# Does the introduction of a cobot change the productivity and posture of the operators in a collaborative task?

**Kévin Bouillet**[1]*, **Sophie Lemonnier**[2], **Fabien Clanche**[3], **Gérome Gauchard**[1,3]

**1** EA 3450 DevAH "Développement, Adaptation et Handicap", Université de Lorraine, Vandœuvre-lès-Nancy, Metz, France, **2** EA 7312 PErSEUs "Psychologie Ergonomique et Sociale pour l'Expérience Utilisateurs", Université de Lorraine, Metz, France, **3** Faculté des Sciences du Sport, Université de Lorraine, Villers-lès-Nancy, Metz, France

* kevin.bouillet@univ-lorraine.fr

**Data Availability Statement:** All files are available from the database: https://osf.io/qe4w7/?view_only=8253c092322348c8a92b290fbdf53799.

## Abstract

Musculoskeletal disorders (MSDs) are the main occupational diseases and are pathologies of multifactorial origin, with posture being one of them. This creates new human-robot collaboration situations that can modify operator behaviors and performance in their task. These changes raise questions about human-robot team performance and operator health. This study aims to understand the consequences of introducing a cobot on work performance, operator posture, and the quality of interactions. It also aims to evaluate the impact of two levels of difficulty in a dual task on these measures. For this purpose, thirty-four participants performed an assembly task in collaboration with a co-worker, either a human or a cobot with two articulated arms. In addition to this motor task, the participants had to perform an auditory task with two levels of difficulty (dual task). They were equipped with seventeen motion capture sensors. The collaborative work was filmed with a camera, and the actions of the participants and co-worker were coded based on the dichotomy of idle and activity. Interactions were coded based on time out, cooperation, and collaboration. The results showed that performance (number of products manufactured) was lower when the participant collaborated with a cobot rather than a human, with also less collaboration and activity time. However, RULA scores were lower—indicating a reduced risk of musculoskeletal disorders—during collaboration with a cobot compared to a human. Despite a decrease in production and a loss of fluidity, likely due to the characteristics of the cobot, working in collaboration with a cobot makes the task safer in terms of the risk of musculoskeletal disorders.

## Introduction

In the annual report of the French Health Insurance on health risks [1], musculoskeletal disorders (MSDs) represent the majority of occupational diseases, accounting for nearly 90% of them. MSDs are pathologies that mainly affect the upper limbs and the back [1]. Therefore, it

**Funding:** The author(s) received no specific funding for this work.

**Competing interests:** The authors have declared that no competing interests exist.

is important to prevent these MSDs. The Institut National de Recherche et de Sécurité (INRS or *National Institute for Research and Safety* in English) has modeled the risk factors for MSDs [2]. MSDs result from multiple dimensions of the work environment, including biomechanical constraints (repetitiveness, posture), work organization, psychosocial factors, and stress. Repetitive tasks in awkward postures are a major risk factor. The posture of the upper limbs and the applied force are related [3,4]. Maximum force can be developed with the shoulder forward at 45°, elbow at 90°, and a neutral position of the wrist and forearm [3]. This posture also allows for better efficiency of force production. The optimal combination of angles depends on the specific task [4]. Deviating from these angles corresponds to awkward postures. Assessing the posture during workers' working time can help predict the occurrence of these disorders. The RULA assessment is generally used to evaluate the risk of developing MSDs based on the operator's posture [5].

Some devices or physical assistance robots aim to lighten the constraints on operators' posture in order to reduce the risk of developing MSDs. These devices, such as collaborative robots (cobots), aim to optimize working conditions for operators and productivity. A human-robot system is described based on workspace, working time, objective, and contacts [6]. From this, human-robot interactions are described as coexistence, cooperation, or collaboration [7], which necessarily differ from a conventional collaboration with a human. For example, communication, roles, stress, posture, or cognitive load may be different, resulting in a change in performance. The introduction of a cobot creates a new interaction, raising questions about the quality of interactions and the impact on the operator's health, including the occurrence of MSDs. This collaboration requires an adaptation of the operator's behavior, as the introduction of the cobot transforms the relationship between their task and themselves, as well as their motor activity, performance, health, and safety [8–10]. The behavior of the cobot influences production and interactions with operators. The programming of the cobot can be based on user preferences (e.g., trajectory or speed) [11], and its coordination (i.e., proactive, reactive, or adaptive) [12], as well as task characteristics (e.g., force to be applied, type of posture and operation, body part used, task complexity), can influence these parameters.

The introduction of a cobot must consider professional constraints. Considering these constraints helps maintain good fluidity in interactions between humans and the cobot, which describes the quality of interactions [13]. The actions of the robot and the human operator can be decomposed through the active and inactive dichotomy [13]. By combining the actions of each, these variables measure the fluidity of interactions, including the inactive time of the robot and the human and the concomitant activity between the two workers [13–16]. As working time is one of the factors contributing to MSDs [17], increasing the active time of operators should increase the risk of developing MSDs.

Furthermore, performing an industrial task can involve simultaneously performing another task, whether physical or cognitive, placing the individual in a dual-task situation. Performing a dual task creates interference and generally degrades performance in one or both of the tasks [18–21]. Increasing the difficulty level of one task degrades performance more than the other [22], as the more difficult task requires more attentional resources [23,24].

Recently, some studies have investigated the impact of introducing a cobot into a task to make it collaborative on production performance and operator health [25,26]. The results showed that RULA scores were lower for operators when working with the collaborative robot compared to working individually. Additionally, production performance was not degraded with the introduction of the cobot into the task. Thus, introducing a cobot into a task appears to be beneficial for the health of operators without reducing production. On the other hand, in certain industrial tasks, human-human collaboration is present, and the constraints applied to one of the co-workers might suggest the possibility of introducing a cobot to lighten the

physical constraints on one of the co-workers. To our knowledge, no study has compared human-human collaboration and human-robot collaboration for the same task in terms of production performance, operator health, or the quality of interactions.

In this regard, the aims of this study were twofold. The first was to compare the number of products manufactured in four minutes, the posture and workload of the operators, as well as the quality of interactions between a human-human system and a human-cobot system performing the same industrial task. The second was to evaluate the influence of two levels of difficulty of a second task on all parameters, by directing attentional resources to a non-contiguous task.

## Material and methods

### Participants

Thirty-four volunteers participated in the study, all of whom were students (13 females and 21 males; aged 22.1±2.0 years). The number of participants was determined using the recommendations of Baguley [27] with a risk α of 0.05 and a power of 1 - β of 0.9. The participants were not familiar with assembly line work or working with a cobot. They had no impairments affecting motor control or attentional behavior, and their vision did not require correction. The Ethics Committee Sud Méditerranée reviewed our application on June 1st (registration number 2021-A00471-40) and stated that "Le CPP n'a pas à émettre un avis éthique sur ce type de recherche qui ne semble pas être une RIPH. Cette étude semble correspondre à des expérimentations en sciences humaines et sociales dans le domaine de la santé". All participants provided written consent prior to their participation.

### Materials and tasks

**Motor Task: Cobot or Human co-worker.** During this experiment, participants performed a motor task in which they had to manufacture products in collaboration with a co-worker on a collaborative working plan (see Fig 1A). The products consisted of a fairing, an SFP product (aluminum product), a cover, three nuts, and three screws (see Fig 1B).

The participants and the co-worker had predefined sub-tasks in the manufacturing process. The co-worker moved the fairing onto the central base, then the participant inserted an SFP product inside it before the co-worker placed a cover on top. The participant inserted a nut

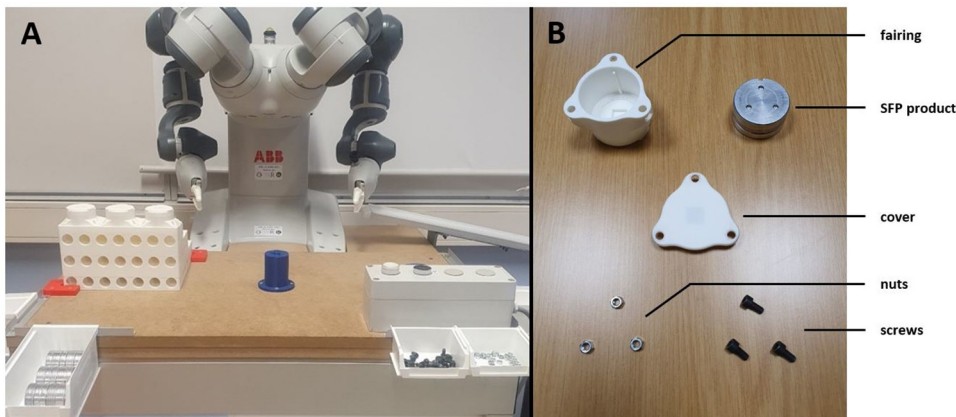

**Fig 1. Working plan and components of the motor task products.** (A) Working plan when a participant works with the YuMi cobot co-worker. (B) Components for manufacturing the products.

into the fairing slot in front of them and then hand-tightened it. The co-worker rotated the product by 120˚ twice, and the participant performed the same operation each time. Once the product was completed, the co-worker moved the product to the unloading ramp, and then the participant and co-worker repeated all these operations to produce the next products. The different manufacturing steps are presented in S1 Appendix.

The working plan consisted of three areas: co-worker area, interaction area, and participant area. Two-centimeter-thick extruded polystyrene plates were used to adjust the height of the working plan so that it was always positioned 5 to 10 cm below the participants' elbow height [28]. They were placed under the participants when necessary.

The aim of this task was to manufacture as many products as possible within four minutes (i.e., the duration of a trial). For each trial, the number of manufactured products was counted, and for the last ongoing product, the number of manufacturing steps was recorded.

As mentioned earlier, the participants had to manufacture products with a co-worker: either a cobot (COB modality) or another human (HUM modality). In the COB modality, the cobot used was a YuMi cobot (ABB Group, Zurich, Switzerland), which consists of two independent arms with seven points of articulation. Both arms are equipped with two grippers at their ends and a vacuum system at the left "wrist." During this collaboration, the production pace was led by the participants. Using a control box, they provided instructions to the cobot for it to perform its next sub-task. The cobot operated at an "automatic" speed, with the speed of arm movements limited to 1.5 m/s.

In the HUM modality, the co-worker was the same for all participants. The co-worker learned their sub-tasks and trained to work at the same speed regardless of the participant's working speed. Like the cobot, the co-worker had reactive coordination, meaning they did not anticipate the participant's actions, so they started their actions only after the participant had finished their sub-task [12], without the participants signaling that it was their turn. To make the HUM modality as similar as possible to the COB modality, verbal communication between the participants and the human co-worker was not allowed. The auditory task, which will be detailed later, controlled for this.

**Auditory task.** The aim of the second task was to place the participants in a dual-task situation to assess the impact of cognitive load without creating interference. Since the main task was visuomotor, this second task was auditory-verbal to avoid using the same modalities [29–31].

The auditory task was similar to the one used in Richer and Lajoie's experiment [24]. Participants listened to a four-minute audio recording of a series of letters (B, D, P, and T) announced randomly. The objective for participants was to count the number of occurrences of a predetermined letter (one of the four letters) in the recording. Participants provided their response (i.e., how many times they heard the predetermined letter) after the four-minute recording. This task had two levels of difficulty based on the inter-stimulus interval (ISI): difficult with a 2-second ISI (2s modality) and easy with a 5-second ISI (5s modality).

The letters were recorded once by a speech therapist, and then MATLAB recordings were generated using a program. Participants listened to the recording using a SoundLink II wireless headset (Bose, Framingham, MA, USA). During the task, participants were not allowed to count on their fingers, maximizing cognitive effort. Additionally, this auditory task prevented verbal communication between the two humans while they were working together.

The success rate for each condition was evaluated. Additionally, for each trial, the absolute error between the participants' response and the correct response was calculated (i.e., how much they were off by). They did not receive feedback on their response.

**Video coding to describe interactions between participants and co-worker.** Each four-minute trial between participants and co-worker was filmed with a QFHD Pro Series camera

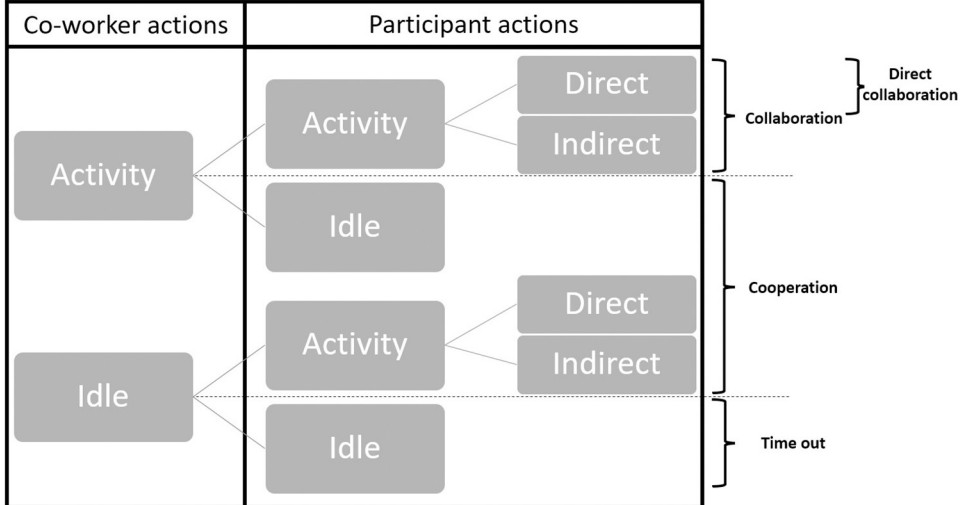

**Fig 2. Action coding and distinction between the different types of interactions.**

at 15Hz (Dahua, Hangzhou, China) mounted on the wall. Then, the video recordings were exported from Smart PSS 1.13.1 software. These videos were imported into Captiv-L7000 2.3.18 software (TEA, Vandœuvre-lès-Nancy, France). Once imported, a coding was created in order to quantify the co-workers' and participants' actions.

Following Hoffmann [13], co-worker's actions and participant's actions were coded according to the "Idle" (i.e., no action on the product) and "Activity" (i.e., action on the product) dichotomy. For participant's actions, there was an additional subdivision, as in Neumann et al.'s article [32], for their "Direct activity" (i.e., motor action on the product at the central base), also known as value-adding work [32,33], and "Indirect activity" (i.e., motor action outside the central base). This coding was represented in Fig 2. All these actions were calculated as a percentage of a trial (i.e., four minutes). Co-worker "Activity" and participants' "Idle" and "Direct activity" were also measured in average duration for each action in seconds (i.e., mean duration of an action).

We propose, drawing from Hoffmann's article [13], a distinction between the different types of interactions based on the actions of the participant and the co-worker (Fig 2). When both were "Idle", there was no interaction, and it was considered as "Time out". When one of them was in "Activity" while the other was in "Idle", it was categorized as "Cooperation". When both were in "Activity", it was classified as "Collaboration", and if the participant was in "Direct activity", it was labeled as "Direct collaboration". These four interactions were expressed as a percentage of a trial.

This division allowed us to analyze the impacts of the cobot and the dual task on all measures (e.g., RULA scores, which will be explained later) according to the type of interaction. It will also enable us to be more precise in the analysis of our data and their interpretation.

**Motion capture to assess the risk of developing MSDs.** Participants were equipped with seventeen motion sensors of MVN Biomech Awinda wireless inertial unit type, recording at 60Hz (Xsens, Enschede, the Netherlands). These sensors were placed on the head, shoulders, sternum, pelvis, arms, forearms, hands, thighs, legs, and feet using scratch strips [34]. The sensors provide positional coordinates in the three planes of space and quaternion orientation. Using a biomechanical model integrated into Xsens MVN Animate Pro software 2021.0.0 and a MATLAB program, joint angle data were calculated in the three axes.

From this data and with a MATLAB program, a RULA assessment per side (i.e., left and right) was conducted. The method for performing the RULA assessment is explained in the article by McAtamney and Corlett [5]. The RULA score, ranging from 1 to 7, was obtained from the joint angles of one of the upper limbs (i.e., shoulder, elbow, and wrist joints), trunk, and neck. Thus, a RULA score was continuously measured.

The average score was calculated by averaging all the scores over the four minutes for each trial. The RULA score was also calculated for different actions of the participants (i.e., "Idle", "Direct activity", and "Indirect activity").

**Experimental process.** Participants performed all the modalities of the two tasks, resulting in four conditions: collaboration with the cobot and a 2-second ISI (COB2s); collaboration with the cobot and a 5-second ISI (COB5s); collaboration with the human and a 2-second ISI (HUM2s); collaboration with the human and a 5-second ISI (HUM5s).

For each condition, they completed three four-minute trials. Participants started either with the two COB conditions followed by the two HUM conditions, or vice versa. Within each block, they began with either three trials in the 2-second modality followed by three trials in the 5-second modality, or the reverse order.

Firstly, participants read an information document about the experiment and signed a consent form. Then, they were equipped with motion sensors on various parts of their body, as explained earlier. Once equipped, participants had to go through the calibration steps for this tool: they stood in an N-pose (standing with both arms at their sides, palms facing their bodies) for two seconds, then walked a seven-step round trip at a "normal" pace and returned to the N-pose for approximately 15 seconds.

Once the equipment and calibration with Xsens were completed, participants went through three learning phases:

First, the motor task as a single task in the COB modality: The different manufacturing steps were verbally explained to the participants, followed by the production of at least six products with the cobot co-worker, with the first two products being manufactured under the author's supervision.

Second, the auditory task as a single task in the 5-second modality: The task and its objective were explained to the participants. They then sat and listened to a learning recording, with a 5-second ISI, for four minutes, focusing on a predetermined letter. After this time, they provided their account and received feedback on their response.

Third, the dual task in the COB5s condition: After learning the two tasks in the single-task condition, participants performed a learning trial in the 5-second condition for four minutes.

When participants were fully equipped and ready to start, they positioned themselves facing the working plan and the co-worker, either a human or a cobot depending on their first condition. They completed the four conditions, with three four-minute trials for each condition, with approximately two minutes of break between two trials to arrange the different elements of the product on the working plan. To start a trial, participants placed their right hand on the central base and did not move it. The author initiated a video recording and a motion capture recording, and then participants quickly moved their right hand to press a remote control on their right side to trigger the cobot co-worker program (they simulated the same action with the human co-worker). Simultaneously, the author triggered the auditory task.

Overall, participants were present on the premises for approximately two and a half hours.

**Data analysis.** For all variables, tests were conducted to compare the four conditions (COB2s, COB5s, HUM2s, and HUM5s), the two modalities of the motor task (with the cobot co-worker versus with the human co-worker), and the two modalities of the auditory task (with a two-second ISI versus a five-second ISI). All statistical tests were performed using the STATISTICA software.

For performance in both tasks, the number of manufactured products (motor task) and the absolute error (auditory task) were measured. The four conditions were compared using a non-parametric repeated measures Friedman's ANOVA. When the test was significant, a Wilcoxon matched pairs test was conducted to observe significant differences between the conditions. Wilcoxon matched pairs tests were also conducted to test the differences between the two modalities of the motor task and auditory task.

For the success rate in the auditory task trials, which is a bimodal qualitative variable (success or fail), a chi-square test was performed to compare the different conditions and modalities. To assess a possible learning effect with the co-worker, the last trial and the first trial of both modalities of the motor task were compared using Wilcoxon matched pairs tests.

For all other variables (such as co-worker and participant's action times, mean times of specific actions, time of different types of interactions, RULA mean scores for each side and RULA mean scores for each side during participant's actions), intra-individual comparisons were made between conditions and modalities to assess differences. Friedman's non-parametric repeated measures ANOVA tests were conducted. When the test was significant, a Wilcoxon matched pairs test was conducted to observe significant differences between the conditions. Wilcoxon matched pairs tests were also conducted to test the differences between the two modalities of the motor task and auditory task.

RULA scores during participants' actions were compared between the three actions (Idle, Direct activity, and Indirect activity) regardless of the condition or modality. They were compared using Friedman's non-parametric repeated measures ANOVA. When the test was significant, a Wilcoxon matched pairs test was conducted to observe significant differences between the participants' actions.

Correlations were made between the three different participants' actions and with the number of manufactured products and RULA scores on both sides to observe relationships between these variables. For significant correlations, a labeling system exists to categorize r values, with $r \leq 0.35$ indicating low correlations, 0.36–0.67 indicating moderate correlations, 0.68–1.00 indicating high correlations, and $r \geq 0.9$ indicating very high correlations [35].

Data that were greater or smaller than the mean plus or minus three standard deviations were considered outliers, which were replaced with mean data for the four conditions or the two modalities [36]. Since comparisons were made using non-parametric tests, the results were expressed as median (interquartile range). The significance level α was set at 0.05.

Detailed results (i.e., mean of the trials of each variable for conditions and modalities of each participant) are available on Open Science Framework: https://osf.io/qe4w7/?view_only=8253c092322348c8a92b290fbdf53799.

## Results

### Tasks performance

**Motor task.** Regarding performance in the motor task, more products were manufactured when participants worked in the HUM modality compared to the COB modality (8.03 (1.22) vs 5.35 (0.82), $p < 0.001$), but there was no difference between the 2s and 5s modalities ($p = 0.657$). The number of products manufactured differed depending on the condition ($\chi^2$ (3) = 82.1, $p < 0.001$). The values and differences for the number of products manufactured in the four conditions are presented in Table 1. Thus, human-human collaboration was more efficient than human-cobot collaboration.

The last trial in the COB modality (5.4 (0.9)) and HUM modality (8.4 (1.2)) was superior to the first trial of the same modalities, respectively (5.2 (0.9) and 7.3 (1.3), $p < 0.001$ for both modalities). Participants continued their learning process during trials of the same modality.

**Table 1. Task performances for the four conditions.**

| Tasks performances | COB2s | COB5s | HUM2s | HUM5s |
|---|---|---|---|---|
| Number of products manufactured | 5.3 (0.6) ‡ | 5.3 (0.9) ‡ | 8.1 (1.3) § | 8.0 (1.4) § |
| Success rate (%) | 12.4 ‡ | 31.0 § | 6.9 ‡ | 31.4 § |
| Absolute error | 4.0 (2.5) ‡ | 1.2 (1.0) § | 2.8 (2.0) # | 1.2 (1.0) § |

COB2s: Collaboration with the cobot and a 2-second ISI. COB5s: Collaboration with the cobot and a 5-second ISI. HUM2s: Collaboration with the human and a 2-second ISI. HUM5s: Collaboration with the human and a 5-second ISI.

‡, § or #: For a line, identical symbols indicate no difference between conditions, while different symbols indicate significant difference between conditions.

**Auditory task.** During the performance of the auditory task, the success rate depended on the level of difficulty ($\chi^2$ (1) = 28.9, p < 0.001), with better success in the 5s modality than in the 2s modality, but there was no difference between the COB and HUM modalities. It also depended on the conditions ($\chi^2$ (3) = 29.8, p < 0.001). The values and differences for the success rate in the four conditions are presented in Table 1. The absolute error in the auditory task did not differ between the COB and HUM modalities (p = 0.289). However, it was higher for the 2s modality than the 5s modality (p < 0.001). This absolute error differed depending on the condition ($\chi^2$ (3) = 51.9, p < 0.001). The values and differences for the absolute error in the four conditions are presented in Table 1. These results confirmed that the modalities of the auditory task corresponded to two levels of difficulty, with the 2s modality being the most difficult.

## Types of actions and interactions

In this section, the results of participants' actions and co-worker's actions are described first, followed by those of the types of interactions.

**Participants' actions.** The participants' idle time was inversely proportional to their activity time, so only the data of their activity are presented here. Since participants' activity is also divided into direct activity and indirect activity, these data are also presented. Participants were more active in the HUM modality than in the COB modality (p < 0.001). Their direct activity and indirect activity were also more significant in the HUM modality than in the COB modality (p < 0.001 for both variables). The results showed that participants' activity time was greater in the 5s modality than in the 2s modality (p = 0.01). No difference was observed between the 5s and 2s modalities for direct activity (p = 0.061) and indirect activity (p = 0.061). For the three variables (i.e., activity, direct activity, and indirect activity times) the results differed according to the condition ($\chi^2$ (3) = 82.5, p < 0.001; $\chi^2$ (3) = 82.1, p < 0.001 and $\chi^2$ (3) = 46.7, p < 0.001, respectively). These times were more significant for the HUM2s and HUM5s conditions compared to the COB2s and COB5s conditions (p < 0.001 for all four comparisons, with significant differences for all three variables). These results are shown in Fig 3. Thus, participants were more active when working with the human co-worker than with the cobot co-worker.

The mean times of participants' idle and direct activity are also calculated. The values and differences for these times are described for the four conditions and presented in Table 2. The results showed that the mean idle time of participants was longer in the COB modality than in the HUM modality (3.11s (0.99) vs. 1.93s (0.56), p < 0.001), while there was no difference between the 2s and 5s modalities (p = 0.638). This time differed depending on the condition ($\chi^2$ (3) = 60.4, p < 0.001). The mean direct activity time of participants did not differ between the COB and HUM modalities (p = 0.590), but this mean time was more significant in the 5s modality than in the 2s modality (p = 0.045). This time did not differ according to the

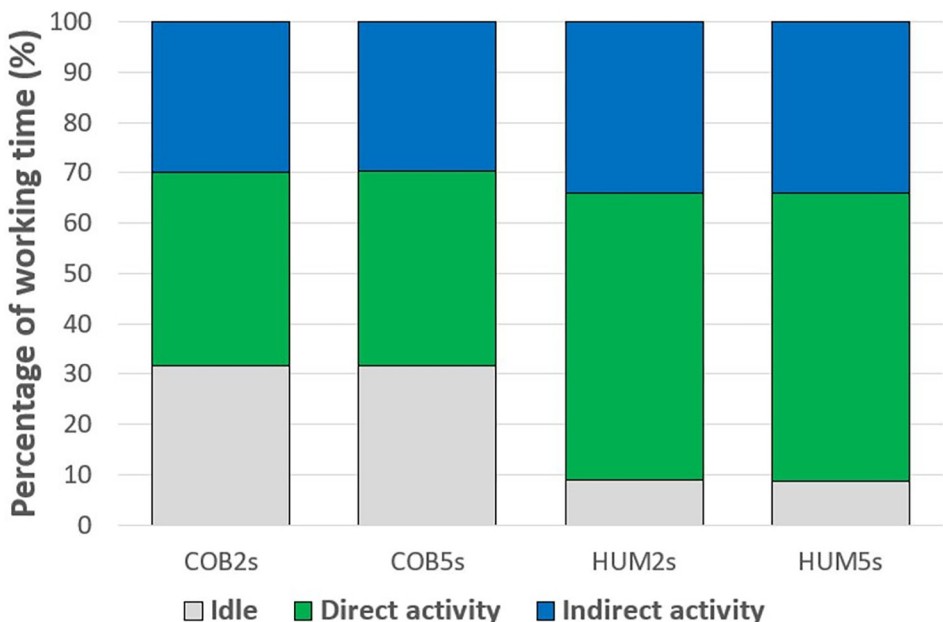

**Fig 3. Distribution of different participants' actions according to the condition.** COB2s: Collaboration with the cobot and a 2-second ISI. COB5s: Collaboration with the cobot and a 5-second ISI. HUM2s: Collaboration with the human and a 2-second ISI. HUM5s: Collaboration with the human and a 5-second ISI. Idle: When participants are in Idle. Direct activity: When participants are in Direct activity. Indirect activity: When participants are in Indirect activity.

condition ($\chi^2$ (3) = 0.4, p = 0.950). Therefore, participants had longer inactive periods with the cobot co-worker than with the human co-worker, and they had longer periods of direct activity with the easiest level of difficulty in the auditory task.

**Co-worker's actions.**   As the co-worker's activity time was inversely proportional to its idle time, only the data of its activity are presented here. The cobot co-worker was more active than the human co-worker (36.1% (9.0) vs. 30.7% (5.1), p < 0.001), while there was no difference in the co-worker's activity time between the 2s and 5s modalities (p = 0.844). The results showed differences in the co-worker's activity time depending on the condition ($\chi^2$ (3) = 62.1, p < 0.001), with greater activity times for the HUM2s and HUM5s conditions compared to the COB2s and COB5s conditions (p < 0.001 for all four comparisons with significant differences).

Additionally, the cobot co-worker's mean activity time was more significant than that of the human co-worker (p < 0.001), but there was no difference in the co-worker's mean activity

**Table 2. Participant mean Idle and Direct activity times and co-worker mean Idle time.**

|  | COB2s | COB5s | HUM2s | HUM5s |
|---|---|---|---|---|
| Participant mean Idle time (s) | 3.3 (0.9) [‡] | 3.2 (1.1) [§] | 1.9 (0.6) [#] | 1.9 (0.5) [#] |
| Participant mean direct activity time (s) | 4.1 (0.8) | 4.1 (0.8) | 4.1 (1.1) | 4.2 (1.0) |
| Co-worker mean Idle time (s) | 3.8 (1.3) [‡] | 3.8 (1.3) [‡] | 2.2 (0.5) [§] | 2.2 (0.6) [§] |

COB2s: Collaboration with the cobot and a 2-second ISI. COB5s: Collaboration with the cobot and a 5-second ISI. HUM2s: Collaboration with the human and a 2-second ISI. HUM5s: Collaboration with the human and a 5-second ISI.

[‡], [§] or [#]: For a line, identical symbols indicate no difference between conditions, while different symbols indicate significant difference between conditions. No symbol indicates that there is no significant difference between the four conditions for a line.

**Table 3. Times of the types of interactions between the participant and the co-worker (% of a trial).**

| Type of interactions | COB2s | COB5s | HUM2s | HUM5s |
|---|---|---|---|---|
| Time out | 9.7 (5.3) ‡ | 9.4 (5.0) ‡ | 0.9 (1.4) § | 1.0 (0.7) § |
| Cooperation | 73.7 (4.3) | 73.3 (4.7) | 74.7 (3.6) | 73.9 (4.4) |
| Collaboration | 17.0 (7.2) ‡ | 17.4 (8.1) ‡ | 24.1 (4.5) § | 25.0 (5.4) § |
| Direct collaboration | 1.9 (2.8) | 3.0 (3.5) | 1.6 (3.3) | 2.0 (4.7) |

COB2s: Collaboration with the cobot and a 2-second ISI. COB5s: Collaboration with the cobot and a 5-second ISI. HUM2s: Collaboration with the human and a 2-second ISI. HUM5s: Collaboration with the human and a 5-second ISI.

‡ or §: For a line, identical symbols indicate no difference between conditions, while different symbols indicate significant difference between conditions. No symbol indicates that there is no significant difference between the four conditions for a line.

time between the 2s and 5s modalities (p = 0.966). This mean time differed between conditions ($\chi^2$ (3) = 62.1, p < 0.001). The values and differences for this time are described for the four conditions and presented in Table 2.

Thus, the cobot co-worker was more active, with longer activities, than the human co-worker, without any interactions with the modalities of the auditory task.

**Interactions between participant and co-worker.** The results showed that there were more "Time out" occurrences in the COB modality than in the HUM modality (p < 0.001), but there was no difference between the 2s and 5s modalities (p = 0.752). The occurrence of "Time out" varied depending on the condition ($\chi^2$ (3) = 72.6, p < 0.001). The results and differences are presented in Table 3. Regarding cooperation time, there was no difference between the COB and HUM modalities (p = 0.114) and between the 2s and 5s modalities (p = 0.256). However, the results showed differences between conditions ($\chi^2$ (3) = 11.5, p = 0.009). The results and differences are presented in Table 3.

Collaboration time was higher in the HUM modality than in the COB modality (24.45% (4.73) vs 17.16% (7.70), p < 0.001). However, the results showed no difference between the COB and HUM modalities for direct collaboration (p = 0.925). The results showed that collaboration time did not differ between the 2s and 5s modalities, whereas direct collaboration time was higher in the 5s modality than in the 2s modality (p = 0.014). Collaboration time varied depending on the condition ($\chi^2$ (3) = 72.6, p < 0.001). These results and differences are presented in Table 3.

Thus, when participants worked with the cobot co-worker, there were more "Time out" occurrences but less collaboration compared to when participants worked with the human co-worker.

## Biomechanical behavior: RULA evaluation

Throughout each trial, a RULA assessment was conducted to continuously evaluate the risk of developing MSDs. A RULA score is a discrete value between 1 and 7, with higher scores indicating a greater risk of developing MSDs. Thus, for each trial, the average RULA score was calculated, as well as the mean RULA scores during different participant actions (i.e., Idle, Direct activity, and Indirect activity). The analysis was performed for both the right and left sides. The results for the mean RULA score are described first, followed by RULA scores for participant actions, and finally, a comparison between the three RULA scores during participant actions, irrespective of the conditions.

**RULA score.** The mean RULA score was higher for participants in the HUM modality (3.61 (0.44) for the right side and 3.95 (0.44) for the left side) compared to the COB modality (3.40 (0.46) for the right side and 3.62 (0.58) for the left side) for both sides (p < 0.001 for both

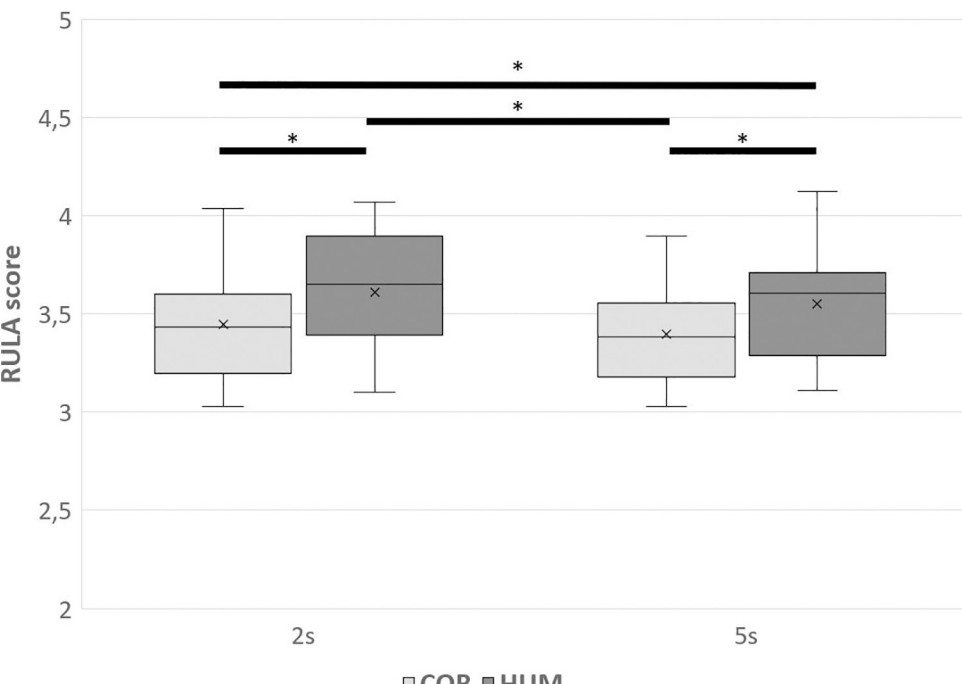

**Fig 4. Mean RULA scores for the right side.** COB: Collaboration with the cobot. HUM: Collaboration with the human. 2s: Two-second ISI. 5s: Five-second ISI. * significant difference between conditions.

sides). A significant difference was observed between the 2s and 5s modalities for the right side, with a higher RULA score for the 2s modality (p = 0.045), but there was no difference for the left side (p = 0.521). The mean RULA score differed depending on the condition for both sides ($\chi^2$ (3) = 27.1, p < 0.001 for the right side and $\chi^2$ (3) = 26.9, p < 0.001 for the left side), with higher scores in the HUM2s and HUM5s conditions compared to COB2s and COB5s (for the left side: p < 0.001 for all comparisons with significant differences; for the right side: Fig 4). Results of the four conditions for the right side are shown in Fig 4. It should be noted that the RULA score was higher for the left side than the right side, regardless of the conditions (p < 0.001).

**RULA score during participant's actions.** When participants were Idle, the RULA score was higher in the HUM modality than in the COB modality (p < 0.001 for the right side and p = 0.011 for the left side). However, there was no difference between the 2s and 5s modalities. The RULA score during Idle differed depending on the condition ($\chi^2$ (3) = 26.7, p < 0.001 for the right side and $\chi^2$ (3) = 14.2, p = 0.003 for the left side). This score was higher in the COB2s and COB5s conditions than in the HUM2s and HUM5s conditions (right side: Table 4; left side: p < 0.001 for HUM2s and COB2s, p = 0.013 for HUM2s and COB5s), except between HUM5s and COB2s (p = 0.052) and for HUM5s and COB5s (p = 0.054) for the left side.

When participants were in Direct activity, the RULA score was higher in the HUM modality than in the COB modality (p < 0.001 for the right side and p = 0.019 for the left side). This score was also higher in the 2s modality than in the 5s modality for the right side (p = 0.032), but there was no difference for the left side (p = 0.397). The RULA score during Direct activity differed depending on the condition ($\chi^2$ (3) = 21.9, p < 0.001 for the right side and $\chi^2$ (3) = 15.1, p < 0.002 for the left side). The results and differences are indicated in Table 4. For the left side, the score was higher for HUM2s compared to the COB2s condition (p = 0.014) and COB5s condition (p = 0.021).

**Table 4. RULA mean scores during the three participants' actions for the right side.**

| Participants' actions | COB2s | COB5s | HUM2s | HUM5s |
|---|---|---|---|---|
| Idle | 3.29 (0.40) ‡ | 3.26 (0.34) ‡ | 3.50 (0.40) § | 3.37 (0.40) § |
| Direct activity | 3.45 (0.54) ‡ | 3.39 (0.53) ‡ | 3.64 (0.55) § | 3.57 (0.54) § |
| Indirect activity | 3.55 (0.32) ‡ | 3.48 (0.35) ‡ | 3.68 (0.51) § | 3.62 (0.45) § |

COB2s: Collaboration with the cobot and a 2-second ISI. COB5s: Collaboration with the cobot and a 5-second ISI. HUM2s: Collaboration with the human and a 2-second ISI. HUM5s: Collaboration with the human and a 5-second ISI.

‡ or §: For a line, identical symbols indicate no difference between conditions, while different symbols indicate significant difference between conditions.

When participants were in Indirect activity, the RULA score was higher in the HUM modality than in the COB modality (p < 0.001 for the right side and p = 0.004 for the left side). No difference between the 2s and 5s modalities was observed. The RULA score during Indirect activity differed according to the conditions ($\chi^2$ (3) = 19.4, p < 0.001 for the right side and $\chi^2$ (3) = 21.9, p < 0.001 for the left side) with a higher score for the COB2s and COB5s conditions than for the HUM2s and HUM5s conditions (for the right side: Table 4; for the left side: p = 0.005 for COB2s and HUM2s, p = 0.001 for COB5s and HUM2s, p = 0.003 for COB5s and HUM5s, and p = 0.012 for COB2s and HUM5s).

**Comparisons between RULA scores during the participants' actions.** For both sides, the RULA scores of the different participants' actions were compared independently of the condition. The results showed that the RULA score was highest when participants were in Direct activity and lowest when they were idle (p < 0.001 for all comparisons and for both sides, see Fig 5 for the right side, the patterns are similar for the left side).

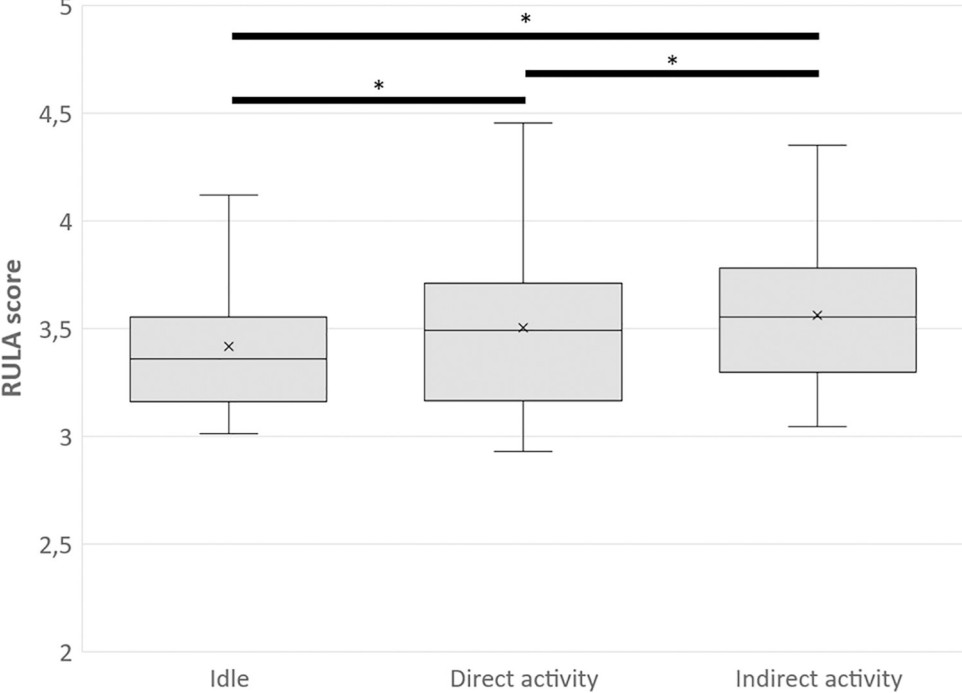

**Fig 5. RULA mean score for the right side during the three participants' actions.** Idle: When participants are in Idle. Direct activity: When participants are in Direct activity. Indirect activity: When participants are in Indirect activity. The comparisons are made independently of the condition. * significant difference between conditions.

## Correlations

Regarding the number of products manufactured and participants' actions, the first one was strongly correlated with their Activity (r = 0.815, p < 0.001), their Direct activity (r = 0.715, p < 0.001), and weakly with the mean time spent in Direct activity (r = -0.219, p = 0.010).

Furthermore, the number of products manufactured was correlated with the collaboration time between participants and their co-worker (r = 0.599, p < 0.001). Participants' Activity was highly correlated with their Direct activity (r = 0.913, p < 0.001), and moderately with the mean time spent in Direct activity (r = 0.248, p = 0.004). Direct activity was weakly correlated with the mean time spent in Direct activity (r = 0.421, p < 0.001).

Low correlations were observed between the number of products manufactured and the mean RULA score for each side (r = 0.2, p = 0.020 for the right side and r = 0.251, p = 0.003 for the left side).

Mean RULA scores were also correlated with participants' actions: their Activity (r = 0.259, p = 0.002 for the right side and r = 0.372, p < 0.001 for the left side), their Direct activity (r = 0.364, p < 0.001 for the right side and r = 0.429, p < 0.001 for the left side), and weakly with the mean time of Direct activity (r = 0.255, p = 0.003 for the right side and r = 0.25, p = 0.003 for the left side).

## Discussion

This study aimed to understand the consequences of collaborative human-robot interactions during assembly work of parts (traditional machining task) on the same working plan.

### Performance at tasks

The results showed that the mode of collaboration has an impact on the production rate. Specifically, the production rate was 50% higher between two humans compared to between a human and the YuMi cobot in this experiment. In previous studies, the introduction of a cobot in an individual task to make it collaborative did not impact productivity [25,26]. Furthermore, participants in those studies benefited from a modification of the working plan and a new distribution of different tasks. In our experiment, the distribution of tasks was similar for participants, so production was not optimized with the collaborative cobot.

In our experiment, the difference in production can be explained by the actual working time of the participants (Activity), as well as the time spent working at the central station (Direct activity), and indirectly by the higher quality of interactions (Collaboration). The more active the participants were, the more products were manufactured. The differences in activity time facing the cobot and the other human can be explained by two factors: production waiting time and the execution speed of the cobot, which was slower than that of the human due to the imposed standards (including limited arm movement speed of 1.5 m/s). Additionally, the human co-worker was asked not to anticipate the end of participants' actions, so the human co-worker started their actions faster than the cobot, particularly due to the need to use the interface to transmit information to the cobot and their reaction time. Despite the reactive coordination of the human co-worker, their reaction time was better. The human co-worker had to exhibit reactive coordination, which is not natural for humans who tend to have proactive coordination [12]. With proactive coordination, performance should be better for collaboration between two humans. Unlike the cobot, humans, with their perceptual abilities, especially gaze, can determine the beginning of their action relative to the end of their co-worker's action, which increases interaction fluidity [14] and collaboration efficiency [37]. When the robot is capable of anticipating human actions, unlike in our experiment, interaction fluidity is improved by reducing waiting time and increasing simultaneous activity time [14], which improves the production rate.

Regarding the auditory task, its modalities corresponded to two levels of difficulty, with the 2s modality being more difficult than the 5s modality [23,24]. Our results showed that the success rate was higher when the ISI was five seconds (31.2% of trials with a correct response to the auditory task) compared to the two-second ISI (9.5%). Moreover, the mean absolute error was lower in the 5s modality, regardless of the mode of collaboration. This auditory task is defined as a continuous task where the two modalities effectively correspond to two levels of difficulty, as in the studies by Richer and Lajoie (2019) and Polskaia and Lajoie (2016) [23,24]. Furthermore, in these two previous studies, their results indicated a comparable performance difference to our experiment but also showed a higher perceived difficulty for the 2s modality. However, their success rates were much higher than in this experiment, but their main task was a postural task that required less attention than the motor task of product manufacturing, allowing for more attentional resources to be allocated to the auditory task.

According to the distinction made by Al-Yahya and colleagues in 2011 [21], there are five types of cognitive tasks in dual-task experiments. Here, the auditory task, which is a cognitive task, corresponds to a working memory task (WMT). WMTs "refer to tasks that require holding information in memory available for processing" [21]. To assess the impact of one task on another, a dual-task cost is evaluated by comparing performance in single-task and dual-task conditions [18,19]. Nearly all dual-task studies with a WMT as a secondary task focused on easy and automated primary tasks, such as postural control with improvement [23,24] or walking with modification of spatiotemporal parameters [21,38–41]. For activities involving the upper limbs, the motor task was generally simple, requiring few degrees of freedom and being repetitive [42,43]. In those studies, results were contradictory, with performance being either degraded or not degraded. Regarding a more complex motor task [44], the automation of it reduced the impact of the secondary task on performance. In our study, the motor task was complex, requiring multiple coordinated movements in space and time in coordination with a co-worker, while the cognitive task (auditory task here) was a WMT. To our knowledge, the results of our experiment align with the first ones comparing the results of a complex motor task with a working memory task.

Thus, the production rate was only impacted by the mode of collaboration and not by the difficulty level of the auditory task. Furthermore, the performance of the auditory task was impacted by difficulty level but not by the mode of collaboration. However, when the auditory task was difficult (i.e., 2-second ISI), the performance of the auditory task was impacted by the mode of collaboration, with degradation in the COB condition (i.e., higher absolute error) compared to the HUM condition. The auditory task required more attentional resources, which appeared to be less available when participants were facing the cobot than when facing the other human. This could indicate an increased need for attentional resources with a cobot. The motor task was not automated in this experiment, as a learning effect was observed regardless of the mode of collaboration, with a greater number of products manufactured in the last trial compared to the first. Due to this lack of automation, the cognitive demand was not negligible for the motor task [45]. The difference in resource requirements for the motor task based on the mode of collaboration would not be sufficient to impact the performance of the auditory task. However, they could be impacted when the amount of attentional resources required for the auditory task is greater [46]. These results raise questions about the amount of attentional resources required in the two modes of collaboration.

## Quality of the interactions

The subtasks were distributed sequentially between the two entities. When a participant finished a subtask, or was in the process of finishing one, it was the co-worker's turn to execute

theirs, and vice versa. In this collaboration, the actions of the cobot or the human co-worker corresponded to the active-inactive dichotomy. Whether it was a cobot or another human, the co-worker was mostly inactive, but even more so when it was a human (69.2% of working time) than when it was with the cobot (62.7% of working time). As for the participants, in the same dichotomy, they were overwhelmingly active, especially when working with a human (91.4% of working time) than with the cobot (69.4% of working time). The distribution of working time was therefore unequal between the two entities of the collaboration. Even with reactive coordination for the human co-worker, the participants' idle time was reduced compared to the cobot co-worker. With proactive coordination, which is more natural for humans, the reduction of this idle time with a human co-worker has already been demonstrated [47].

Focusing more specifically on the Direct activity and Indirect activity dichotomy, participants were also more active at the central base, directly on the product, when facing the cobot compared to the other human (38.1% of working time vs. 56.8% respectively). However, the mode of collaboration had no effect on the average duration of a direct activity. Thus, participants took the same amount of time to perform their subtasks at the central base. Their execution speed was not affected either by the presence of the robot or by the speed of action of the co-worker, which was faster when it was a human. This speed could be influenced by the co-worker's speed [48], a phenomenon known as motor contagion. This has been observed in human-human interactions [49–51] as well as in human-robot interactions [49,52], although there have been few studies in this area. The increase in activity time at the central base was therefore linked to the number of actions participants performed during their working time. In fact, this direct activity time was correlated with the number of products manufactured.

A cobot, as a collaborative robot, differs from a traditional robot in its ability to collaborate with a human, being close to them in terms of safety [7]. In this work situation, the combination of actions from each entity [13] distinguishes two types of interactions between the two entities: cooperation, where one worker acts while the other waits, and collaboration, where both act simultaneously. The objective of this study was to compare the collaboration times with a cobot co-worker and a human co-worker. The results showed a longer collaboration time when facing the cobot compared to facing the other human, which might explain the increase in production, as the number of products manufactured and the collaboration time were correlated. Changing the cobot's coordination to be proactive [12] would enhance this hybrid collaboration to improve fluency and thus increase productivity [14,37]. Despite the distinction made in this study between cooperation and collaboration, interactions between participants and the co-worker could be considered as collaboration according to certain definitions. Indeed, according to Hentout and colleagues [7], collaboration corresponds to the accomplishment of a complex task with direct physical interactions or without contact, which was the case in this complex motor task due to the distribution of subtasks and the necessary coordination between participants and the co-worker, with constant proximity between the two throughout the assembly process.

## Risk of developing MSDs

An ongoing RULA assessment was conducted to quantify the risk of developing MSDs. The RULA score corresponds to a discrete value between one and seven, where the higher the score, the greater the risk. The average score was lower when the participant worked with the cobot coworker compared to the human coworker, with a reduction of 5.0% for the right side and 5.7% for the left side. A decrease in score was also observed when introducing a cobot in an individual work situation in a recent case study [25]. However, with the introduction of the cobot, an ergonomic study was conducted to optimize the operator's posture and reduce this risk.

RULA evaluations use scores to measure the risk of developing MSDs. They utilize optimal angles of upper limb joints as a lower score for articulation, and as one deviates from these angles, the scores increase [3,4]. Thus, the lower RULA scores observed with the cobot indicate that muscular activation could be reduced when working with the cobot coworker compared to the human coworker, thereby reducing the risk of developing MSDs.

The RULA assessment indicated that the risk of developing MSDs was higher when the participants were active, and even more so when they were working at the central base. When participants were inactive, despite the non-value adding activity for the products, it served as a recovery time. Delaying this rest time for the workers implies a higher risk for MSDs [53]. However, regardless of the participants' actions (i.e., inactivity, direct activity, or indirect activity), the RULA score was higher with the human coworker. This score seemed surprising when the participants were inactive, as they were not doing anything and were in a waiting situation. The working plan had been designed to be at the same height for both modes of collaboration, respecting the best ergonomic height [28]. However, their average idle times differed depending on the coworker, being shorter when facing a human compared to the cobot. The idle time can represent a rest period for participants. Since the rest time was shorter with the human coworker, participants had less time and might be ready for the next subtask, indicating they were not in a state of rest.

However, it should be noted that this score, for both sides, was slightly correlated with the number of products. Thus, the RULA score and the activity time might increase when production performance increases. Future studies could investigate the relationship between these variables. The production rate was determined by the participants, and they were free to work at their own pace without time constraints or objectives. But what would happen to this score if a specific cadence was imposed? Or what if the participant is no longer the leader and is guided by the cobot? Further studies could explore these questions, especially when the cadence is controlled at a different speed by the cobot rather than the operator.

**Study limitation: A laboratory study.** One major limitation of this study is that the experiment did not take place in a fully ecological context. Indeed, the experiment was conducted in a laboratory with participants who were not experienced in assembly line work or working with a cobot.

Despite choosing a collaborative work situation that required human presence [54] and high-level interactions between workers [7], the task was performed in a laboratory without a real performance objective. Thus, the behaviors for the participants were not the same as for operators working in a production line [55].

From a health and posture perspective, the participants in our study were young and did not match the typical profile of people affected by musculoskeletal disorders (TMS): individuals aged between 40 and 60 with experience in repetitive tasks [1]. Furthermore, the attitude of a participant in our study differs from that of an industry worker when interacting with a cobot [56], particularly due to the operator's experience influencing their behavior.

Finally, posture studies were conducted on short-duration recordings. In total, the participants worked for less than an hour, while the onset of TMS occurs over the long term with the repetition of movements in awkward postures [2]. Thus, the risks are calculated if the person repeats the same task continuously for much longer periods (i.e., several hours and several days), without considering certain organizational factors such as job rotation during a day [57].

Therefore, it would be interesting to transpose this study to a population of operators performing a task in interaction with a cobot in a real collaborative work situation over longer durations to assess production performance and the long-term impact on the operators' health.

## Conclusion

The introduction of the cobot co-worker reduces productivity compared to a human co-worker in this collaborative work situation. This decrease in performance is linked to a deterioration in the fluidity of the interaction, with more waiting time at the expense of working time on the product or collaboration between the two workers. However, the risk of developing MSDs is reduced with the presence of the cobot for operators in this collaborative situation. Despite the absence of an impact of the auditory task, the second task here, on production performance, increasing the difficulty of the second task reduces the amount of attentional resources and degrades the fluidity of the interaction, thereby increasing the risk of developing MSDs. Therefore, to evaluate an operator in a given workstation, it is necessary to consider all tasks, both motor and cognitive, and not just performance in their specific task, but also all human factors that could be influenced by the introduction of a cobot and the new system. Before introducing a cobot in a specific work situation, it is necessary to quantify the gains and costs of this introduction on production performance and operator factors.

## Supporting information

**S1 Appendix. Different steps to manufacture products.** Here a participant is working with the cobot co-worker. A- Participant press the button on the transmitter; B- Participant inserts an SFP product into the fairing; C- He inserts the first nut; D- Participant screws for the second time; E- He screws for the third time; F- Cobot co-worker evacuates the product and brings the next fairing to the central base. The individual in this manuscript has given written informed consent (as outlined in PLOS consent form) to publish these case details. (TIF)

## Acknowledgments

We are grateful to AIP Lorrain, in particular its director Muriel Lombard, to welcome us in its premises. We thank also Christian Ulrich Tchounke Lonang and Hugo Portejoie for their technical support and the speech language therapist Amélie Dumont for the recording of the letters of the auditory task.

## Author Contributions

**Conceptualization:** Kévin Bouillet, Sophie Lemonnier, Fabien Clanche, Gérome Gauchard.

**Data curation:** Kévin Bouillet.

**Formal analysis:** Kévin Bouillet.

**Funding acquisition:** Kévin Bouillet.

**Investigation:** Kévin Bouillet.

**Methodology:** Kévin Bouillet, Sophie Lemonnier, Fabien Clanche.

**Project administration:** Sophie Lemonnier, Gérome Gauchard.

**Resources:** Kévin Bouillet, Sophie Lemonnier, Gérome Gauchard.

**Software:** Kévin Bouillet, Fabien Clanche.

**Supervision:** Kévin Bouillet, Sophie Lemonnier, Gérome Gauchard.

**Validation:** Kévin Bouillet, Sophie Lemonnier.

**Visualization:** Kévin Bouillet, Fabien Clanche.

**Writing – original draft:** Kévin Bouillet.

**Writing – review & editing:** Sophie Lemonnier, Gérome Gauchard.

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
