## [Decision Letter · Decision Letter 0]

27 Feb 2023

PONE-D-22-15982Does the introduction of a cobot change the posture and the quality of interaction during a collaborative task at work?PLOS ONE

Dear Dr Bouillet,

Thank you for submitting your manuscript to PLOS ONE. After careful consideration, we feel that it has merit but does not fully meet PLOS ONE’s publication criteria as it currently stands. Therefore, we invite you to submit a revised version of the manuscript that addresses the points raised during the review process.

We look forward to receiving your revised manuscript.

Kind regards,

Pooya Sareh, PhD (Cantab.), FHEA, FRSA

Academic Editor

PLOS ONE

Journal Requirements:

3. We note that Figure S1 includes an image of a [patient / participant / in the study].

Reviewers' comments:

Reviewer's Responses to Questions

**Comments to the Author**

1. Is the manuscript technically sound, and do the data support the conclusions?

Reviewer #1: Yes

Reviewer #2: Partly

Reviewer #3: Yes

Reviewer #4: Yes

Reviewer #5: Partly

2. Has the statistical analysis been performed appropriately and rigorously? 

Reviewer #1: Yes

Reviewer #2: Yes

Reviewer #3: Yes

Reviewer #4: Yes

Reviewer #5: Yes

3. Have the authors made all data underlying the findings in their manuscript fully available?

Reviewer #1: Yes

Reviewer #2: Yes

Reviewer #3: Yes

Reviewer #4: Yes

Reviewer #5: Yes

4. Is the manuscript presented in an intelligible fashion and written in standard English?

Reviewer #1: Yes

Reviewer #2: Yes

Reviewer #3: Yes

Reviewer #4: Yes

Reviewer #5: Yes

5. Review Comments to the Author

Reviewer #1: This is an interesting study on the use of a cobot during a collaborative task. The manuscript is well written, the methods and results are clearly described. The conclusion is supported by the results of the study.

I have small comments:

1. The figures have low resolution;

2. Describe the characteristics of the participants (sex, experience and familiarity with the task);

3. Check lines 313-314;

4. Exclude the term "tendency to significance";

5. Avoid distinguishing P<0.05; P<0.01 and P<0.001.

Reviewer #2: Comments:

1. some works are exploiting user's preference to tune the robot behavior or task. These works can be mentioned in the introduction of the paper, such as [1];

2. questionnaires for the perceived usefulness, effort, etc., could have been provided;

3. check the English.it would have been interesting to perform the same task with different (bigger) robots (e.g., a UR10).

[1] Roveda, Loris, et al. "Pairwise preferences-based optimization of a path-based velocity planner in robotic sealing tasks." IEEE Robotics and Automation Letters 6.4 (2021): 6632-6639.

Reviewer #3: Thanks for giving me the opportunity for reviewing this well-carried out study. However, I have following suggestions as pointed out below:

1. The interaction between workers is described properly. However, any work (especially manual work) depends upon several factors such as type of posture, force (grip, push, pull, etc.), body part used (single or multiple), type of operation (manual or mechanical), experience of workers, complexity of the operation, etc.? I think these factors need to be reflected in literature and discussion section of the paper appropriately.

2. The author(s) seem to left out this important link of theory/model in relation to the issue being examined. Perhaps theories/models such as Karsh (2006), Sauter and Swanson (1996), and other relevant models/theories should be included in the appropriate sections to highlight the importance of the theories/models in guiding the study.

3. The results are of some interest but the authors could elaborate further on the practical implications of their findings in the discussion section.

4. Some papers related to posture and forces should also be cited and reviewed, such as: https://doi.org/10.1080/10803548.2014.11077039, https://doi.org/10.1080/10803548.2018.1501972, https://doi.org/10.1016/j.ergon.2004.07.002, https://doi.org/10.1080/10803548.2022.2041300

Reviewer #4: The paper studies the use of a collaborative robot (cobot) during a collaborative work of type assembly of parts on the number of products manufactured in four minutes, operators’ posture and quality of the interactions. They compared a motor task and an auditory task assisted by either the cobot or a human. The subject is interesting and relevant for the research community, however there are a number of aspects that the authors need to address before the paper can be published.

The title of the paper is misleading since the authors are analyzing the “posture and quality” of the interaction but rather the level or quantifying the length of interactions of the participant with the cobot, and its effect on activity and RULA scores. I recommend that the authors change the title to clarify their objective.

The authors should indicate the choice in sample size, since they indicate that 34 volunteers were the number. I would expect a pilot analysis and with that determining the sample size. Please look at the work by Baguley (2004). This is a very important point since if not addressed adequately it can invalidate the results obtained.

I do not understand the introduction of the auditory task in the experiment. What´s the purpose? What should be the interaction with the motor task? If the role is to evaluate cognitive load, shouldn’t the authors increase the number of tasks and their complexity to the participants?

The video motion categorization is interesting along with the use of sensors. Also, the experiment structure is also interesting, but it is not clear how the cobot assists in the process and if a longer interaction (longer than 4 minutes) or learning curves would benefit the cobot. The authors should eliminate the part where they indicate the implementation of the Tobii Pro Glasses, sinceit does not add to the paper.

The results are interesting however the production results are shown in aggregate (For all 4 minutes). It would be interesting to show if there is any time effect or learning curve between the interaction of the cobot and the human, and analyze if the production rate is stable along the time period of the experiment. In the discussion, the production difference is attributed mostly to the working time, but there could be an interaction learning curve between the human and the cobot.

The results and discussion of the auditory tasks do not add to the main focus of the paper and I recommend to be taken out. If the cognitive load is a factor that is to be analyzed, I recommend that the complexity or number of tasks should be increased.

The chapter related the quality of interactions really looks at the amount of interaction, this should be addressed.

The results regarding the MSDs indicate that the cobot has less score. However, since the amount of time active and productive rate was more for the human than the cobot that could be an explanation. The authors could control for the number of units produced and compare the rula scores for equal number of units produced.

Baguley, T. (2004). Understanding statistical power in the context of applied research. Applied ergonomics, 35(2), 73-80.

Reviewer #5: The paper aims to present part of the results of extensive and complex research. However, the authors failed to highlight the segment that is in focus and present it in an appropriate manner. This resulted in an overly extensive paper that is, in certain parts, confusing, imprecise and difficult to follow and read. In addition, the conclusions that the authors generate (primarily related to productivity parameters) refer to the design of a specific experiment and cannot in any way represent conclusions of a general nature.

The recommendation to the authors is to shorten the paper and to focus as much as possible on the defined topic

6. PLOS authors have the option to publish the peer review history of their article (what does this mean?). If published, this will include your full peer review and any attached files.

Reviewer #1: No

Reviewer #2: No

Reviewer #3: No

Reviewer #4: No

Reviewer #5: No

---

## [Author Response · Author response to Decision Letter 0]

29 Mar 2023

Response to Reviewer #1

- General comment:

Reviewer’s comment:

This is an interesting study on the use of a cobot during a collaborative task. The manuscript is well written, the methods and results are clearly described. The conclusion is supported by the results of the study.

Author’s comment:

Thank you for your comment.

- Comment 1:

Reviewer’s comment:

The figures have low resolution.

Author’s comment:

We improved resolutions of the figures. Moreover, figures 4 and 6 were replaced with tables 2 and 4.

- Comment 2:

Reviewer’s comment:

Describe the characteristics of the participants (sex, experience and familiarity with the task.

Author’s comment:

Characteristics of the participants are added in the Material and methods sections (p.5).

- Comment 3:

Reviewer’s comment:

Exclude the term "tendency to significance".

Author’s comment:

We excluded the notion of tendency to significance all time it appeared. In these cases, we indicated that it was not different between the conditions.

- Comment 4:

Reviewer’s comment:

Avoid distinguishing P<0.05; P<0.01 and P<0.001.

Author’s comment:

In the text, we respected PLOS ONE style for P-values: “Report exact p-values for all values greater than or equal to 0.001. P-values less than 0.001 may be expressed as p < 0.001, or as exponentials in studies of genetic associations.” In the figures, we deleted the difference between the three P-values. Differences were shown only with * for significant difference between two conditions.

Response to Reviewer #2

- Comment 1:

Reviewer’s comment:

Some works are exploiting user's preference to tune the robot behavior or task. These works can be mentioned in the introduction of the paper, such as [1].

Author’s comment:

Some works are exploiting it, but in our work, we imposed the robot behavior and the task. We added sentences mentioning these works, as the one recommended by the reviewer, but also Huang and collaborators work in 2015 ([12 now] cited p.7 in methods, p.25 & 29 in Discussion) about robot behavior which one is recommended for a better HRI, especially with production (Introduction section, p.4).

- Comment 2:

Reviewer’s comment:

Questionnaires for the perceived usefulness, effort, etc., could have been provided.

Author’s comment:

We agree with the reviewer that all human factors were not considered in this study. Concerning the MSDs, based on INRS’s model, we mainly focused on biomechanical solicitations (especially with posture and RULA evaluation) and also on workload (with the auditory task). We also focused on productivity and interactions to compare the human-human system and the human-cobot system. Other human factors as perceived usefulness, effort or stress could be measured. As the population here was students and not operators, perceived usefulness was not measured as the expectations of the students would be different than those of real operators. The other dimensions could be evaluated here but we focused on the four dimensions (productivity, interactions, posture and workload). Further studies could interest on these dimensions, with an operators’ population to evaluate the perceived usefulness.

- Comment 3:

Reviewer’s comment:

Check the English.

Author’s comment:

We checked the English and we improved it in the text.

- Comment 4:

Reviewer’s comment:

It would have been interesting to perform the same task with different (bigger) robots (e.g., a UR10).

Author’s comment:

We agree that it could be interesting to perform a similar experimental process with a different cobot (as you mentioned an UR10 or another one). However, adding another cobot to compare the effects of the type of cobot would lighten up the experiment. First, to compare these repercussions by different cobots was not one of our aims, we aimed to compare a human-human system and a human-cobot system. Second, that’s why we do not generalize the results to all types of human-cobot interactions and we use “the co-worker cobot” rather than “a co-worker cobot” (Conclusion section, p.32).

Response to Reviewer #3

- Comment 1:

Reviewer’s comment:

The interaction between workers is described properly. However, any work (especially manual work) depends upon several factors such as type of posture, force (grip, push, pull, etc.), body part used (single or multiple), type of operation (manual or mechanical), experience of workers, complexity of the operation, etc.? I think these factors need to be reflected in literature and discussion section of the paper appropriately.

Author’s comment:

These different factors influence the interaction between the workers (added in Introduction section, p.4) In this experiment, we place us in a manual task which not require lot of force, as subtasks were taken and placed of lightweight components and screwing, using both upper limbs. These factors, and the complexity of operation are described in the article (Motor task sub-section in Material & methods section, p.6-7). The experience of workers is described (Participants sub-section in Material & methods section, p.6). In the discussion section, we explained that the results concerned this kind of task with this cobot (Conclusion section, p.32).

- Comment 2:

Reviewer’s comment:

The author(s) seem to left out this important link of theory/model in relation to the issue being examined. Perhaps theories/models such as Karsh (2006), Sauter and Swanson (1996), and other relevant models/theories should be included in the appropriate sections to highlight the importance of the theories/models in guiding the study.

Author’s comment:

A paragraph is added in the introduction (p.3) to talk about MSDs with a model. We choose the one from INRS (Institut National de Recherche et de Sécurité / National Institute for Research and Security). In this model ([3] Uguen C, Sablon S, Carballeda G. Intégration de la préparation physique dans l’approche ergonomique : quel dispositif pour quel(s) objectif(s) ? Activités. 2018;15: 1–34. doi:10.4000/activites.3537), MSDs are the results of multiple dimensions of the working environment (biomechanical solicitations, working organization, psychosocial factors, stress). Biomechanical solicitations (with posture, repetitiveness as example) are the main factor in this model, that’s why we focus mainly on the participants’ posture in this work. Other factors also have an important role in the appearance of MSDs. Thus, we add a second task to compare different mental workload, as a higher workload is a factor of MSDs.

- Comment 3:

Reviewer’s comment:

The results are of some interest but the authors could elaborate further on the practical implications of their findings in the discussion section.

Author’s comment:

It was not the aim of this experiment which was not a practical case. But we agree that further studies could me more ecological (real situation with real operators as population) in order to make practical implications of the results. We added a sub-section in the discussion section, p.31.

- Comment 4:

Reviewer’s comment:

Some papers related to posture and forces should also be cited and reviewed.

Author’s comment:

In the Introduction section, we add a paragraph to talk about the link between posture and force. We add that some posture allows a higher production of force and also a higher efficiency of muscle to produce force (Introduction section, p.3). In the Discussion section, we remind this. We also add that this task did not require much force. Thus, better upper limb posture (defined with RULA and similar than in these studies given by the reviewer) would decrease the risk of MSDs.

Response to Reviewer #4

- Comment 1:

Reviewer’s comment:

The title of the paper is misleading since the authors are analyzing the “posture and quality” of the interaction but rather the level or quantifying the length of interactions of the participant with the cobot, and its effect on activity and RULA scores. I recommend that the authors change the title to clarify their objective.

Author’s comment:

As productivity and posture are the two mains dimensions in the conclusion, and the quality of the interactions are linked to these two dimensions, we changed the title to highlight these two dimensions. The new title is “Does the introduction of a cobot change the productivity and posture of the operator in a collaborative task?” (p.1).

- Comment 2:

Reviewer’s comment:

The authors should indicate the choice in sample size, since they indicate that 34 volunteers were the number. I would expect a pilot analysis and with that determining the sample size. Please look at the work by Baguley (2004). This is a very important point since if not addressed adequately it can invalidate the results obtained.

Author’s comment:

Before starting the experiment, a pilot analysis was conducted with eight people. To determine the sample size, we measure the number of products manufactured by these eight people realizing two trials similar than those in COB5s and HUM5s. The difference of means was 2.4 with a common standard deviation was about 1.4. With an α risk of 0.05 and a power 1 – β of 0.9 to determine the sample size, it was recommended to have at least sixteen participants at all. For safety reasons, it was decided to double this number. We had this part in Participants sub-section in Material and methods, p.5-6).

- Comment 3:

Reviewer’s comment:

I do not understand the introduction of the auditory task in the experiment. What´s the purpose? What should be the interaction with the motor task? If the role is to evaluate cognitive load, shouldn’t the authors increase the number of tasks and their complexity to the participants.

Author’s comment:

According to the INRS’s model about MSDs ([3] Uguen C, Sablon S, Carballeda G. Intégration de la préparation physique dans l’approche ergonomique : quel dispositif pour quel(s) objectif(s) ? Activités. 2018;15: 1–34. doi:10.4000/activites.3537), the cognitive load is a factor of MSD’s occurrence. Adding the auditory task in this experiment allowed to increase the number of tasks to perform for participants. Moreover, there were two levels of difficulty in this study to compare two cognitive loads.

- Comment 4:

Reviewer’s comment:

The video motion categorization is interesting along with the use of sensors. Also, the experiment structure is also interesting, but it is not clear how the cobot assists in the process and if a longer interaction (longer than 4 minutes) or learning curves would benefit the cobot.

Author’s comment:

We agree with this comment that a longer interaction could be interesting to evaluate the cobot support with time. First, we observed that learning of the task was not fully completed (with both co-worker) (Tasks performance sub-section in Results section, p.15). It could be interesting to complete the learning phase to obtain a stabilization of the performance. To need this, a longer learning phase would be necessary and would extend the experimentation time (A theorical method to apply to a more ecological context sub-section in Discussion, p.31).

- Comment 5:

Reviewer’s comment:

The authors should eliminate the part where they indicate the implementation of the Tobii Pro Glasses, since it does not add to the paper.

Author’s comment:

We understand the comment and we delete the part where we indicate the implementation of the Tobii Pro Glasses.

- Comment 6:

Reviewer’s comment:

The results are interesting however the production results are shown in aggregate (For all 4 minutes). It would be interesting to show if there is any time effect or learning curve between the interaction of the cobot and the human, and analyze if the production rate is stable along the time period of the experiment. In the discussion, the production difference is attributed mostly to the working time, but there could be an interaction learning curve between the human and the cobot.

Author’s comment:

I understand that you suggested analyzing the evolution of interactions and production during the same trial. We could have measured the time to manufacture a product during the trial and assessed whether it changed over time. 

First, we were interested in this question of learning during the experiment, but not during a trial, but rather during different trials with the same co-worker. To do this, we compared the first and last trials in COB modality (with the cobot co-worker) and in HUM modality (with the human co-worker) only for production (Data analysis sub-section, p.13). We observed a learning curve as the production during the last trial with the co-worker was higher than the production during the first trial (Tasks performance sub-section in Results section, p.15);

Second, to avoid any learning or fatigue effects to compare the conditions, two contrabalancements were made (Experimental process sub-section, p.11);

Moreover, we found (Correlations sub-section in Results section, p.23) that the number of products manufactured was highly correlated with the participants’ Activity time (r = 0.815, p < 0.001) and Direct activity time (r = 0.715, p < 0.001). That’s why we attributed the production difference mostly to the working time. We also found (Correlations sub-section in Results section, p.23) that the number of products manufactured was correlated with the Collaboration time (r = 0.599, p < 0.001) to support the fact that the production was linked with the interactions between the participant and the co-worker;

Thus, despite the learning curve observed with both co-workers, contrabalancements avoided to biases the results. A product analysis could be made to evaluate the learning curve product by product, but we preferred to analyze this learning curve trial by trial. The production difference was attributed mostly to the working time, but also to the higher collaborations with the human than with the cobot co-worker (Performance at tasks sub-section in Discussion section, p.25; Conclusion, p.32)

- Comment 7:

Reviewer’s comment:

The results and discussion of the auditory tasks do not add to the main focus of the paper and I recommend to be taken out. If the cognitive load is a factor that is to be analyzed, I recommend that the complexity or number of tasks should be increased.

Author’s comment:

Results of the auditory tasks allowed us to evaluate the cognitive load. In a dual task context, the decrease in performance in the second task show that the cognitive load is more important, and here the attentional resources available to perform it. The number of tasks was increased with the second task (the auditory task).

- Comment 8:

Reviewer’s comment:

The chapter related the quality of interactions really looks at the amount of interaction, this should be addressed.

Author’s comment:

We understand the comment. However, in the literature [14 Hoffman G. Evaluating fluency in human–robot collaboration. IEEE Trans Hum Mach Syst. 2019;49: 209–218. doi:10.1109/THMS.2019.2904558] they defined this chapter as “quality of interaction”. That’s why we use this terminology.

- Comment 9:

Reviewer’s comment:

The results regarding the MSDs indicate that the cobot has less score. However, since the amount of time active and productive rate was more for the human than the cobot that could be an explanation. The authors could control for the number of units produced and compare the RULA scores for equal number of units produced.

Author’s comment:

We fully agree that the amount of time active and production rate was more important for HUM than COB and this difference could be an explanation of the differences of RULA scores. Thus, we aware that it needs to control the number of units produced to compare RULA scores. We calculate the correlation between the number of products manufactured and RULA scores (Correlations sub-section in Results section, p.23) and we found low correlations between these measures (r = 0.2, p = 0.020 for the right side and r = 0.251, p = 0.003 for the left side). That’s why we do not normalize RULA scores according to the number of products manufactured. But to allow a more precise comparison, we calculate RULA scores for the three different participant’s actions RULA scores during participant’s actions sub-section in Results section, p.21-22). With these measures, we weight RULA scores according to the actions and not the production. We found that RULA scores were higher during participant’s Direct activity and participants were more in Direct activity with the human co-worker than with the cobot co-worker. Thus, RULA scores could be influence by these increases. However, RULA scores were more important with the human co-worker than with the cobot co-worker during the three participant’s actions. Thus, higher RULA scores were higher with the human co-worker because the increase of participant’s Direct activity and because higher RULA scores for all participant’s actions.

Response to Reviewer #5

- General comment:

Reviewer’s comment:

The authors failed to highlight the segment that is in focus and present it in an appropriate manner. This resulted in an overly extensive paper that is, in certain parts, confusing, imprecise and difficult to follow and read.

Author’s comment:

We tried to better define the aims of the study (Introduction section, p.5). We hope that the paper is with greater clarity.

We reduce the Results section changing some texts explaining results of conditions in two Tables (Tasks performance sub-section in Results section, p.15; Types of actions and interactions sub-section in Results section, p.17-18) in order to simplify and make the results more readable.

- Comment 1:

Reviewer’s comment:

The conclusions that the authors generate (primarily related to productivity parameters) refer to the design of a specific experiment and cannot in any way represent conclusions of a general nature. The recommendation to the authors is to shorten the paper and to focus as much as possible on the defined topic.

Author’s comment:

Main results concern this variable. In the conclusion, we talked about all variables, with an accent on productivity because results were very significant. In the conclusion section, p.31, we change in order to less generalize all our results (from “a collaborative work situation” to “this collaborative work situation” for example).

---

## [Decision Letter · Decision Letter 1]

22 May 2023

PONE-D-22-15982R1Does the introduction of a cobot change the productivity and posture of the operators in a collaborative task?PLOS ONE

Dear Dr. Bouillet,

Thank you for submitting your manuscript to PLOS ONE. After careful consideration, we feel that it has merit but does not fully meet PLOS ONE’s publication criteria as it currently stands. Therefore, we invite you to submit a revised version of the manuscript that addresses the points raised during the review process.

We look forward to receiving your revised manuscript.

Kind regards,

Pooya Sareh, PhD

Academic Editor

PLOS ONE

Journal Requirements:

Additional Editor Comments:

Reviewer#4 has made comments which require you to make "minor revisions" to your manuscript. Therefore, I would like to invite you to make minor revisions to your manuscript based on the reviewer's comments.

Reviewers' comments:

Reviewer's Responses to Questions

**Comments to the Author**

1. If the authors have adequately addressed your comments raised in a previous round of review and you feel that this manuscript is now acceptable for publication, you may indicate that here to bypass the “Comments to the Author” section, enter your conflict of interest statement in the “Confidential to Editor” section, and submit your "Accept" recommendation.

Reviewer #2: All comments have been addressed

Reviewer #3: All comments have been addressed

Reviewer #4: (No Response)

2. Is the manuscript technically sound, and do the data support the conclusions?

Reviewer #2: Partly

Reviewer #3: Yes

Reviewer #4: Yes

3. Has the statistical analysis been performed appropriately and rigorously? 

Reviewer #2: Yes

Reviewer #3: Yes

Reviewer #4: Yes

4. Have the authors made all data underlying the findings in their manuscript fully available?

Reviewer #2: Yes

Reviewer #3: Yes

Reviewer #4: Yes

5. Is the manuscript presented in an intelligible fashion and written in standard English?

Reviewer #2: Yes

Reviewer #3: Yes

Reviewer #4: No

6. Review Comments to the Author

Reviewer #2: The paper can be now accepted. All the concerns have been properly addressed in the revised paper and in the reply to review file.

Reviewer #3: Thanks for making changes. All the comments raised by reviewer are addressed by the authors in this version.

Reviewer #4: The paper studies the use of a collaborative robot (cobot) during a collaborative work of type assembly of parts on the number of products manufactured in four minutes, operators’ posture, and quality of the interactions. The authors have addressed my concerns in the content of the paper. Some aspects arise from the corrected version.

The authors should send the paper to a native editor to edit the English.

In Table 1, it’s not clear what the authors mean by “a the condition with the highest value.”, “b the condition with lower value than a.”… This is not clear at all. The same is in table 2, 3 and 4.

The added chapter “A theorical method to apply to a more ecological context” does add nothing significant to the paper and only points out one limitation. Instead, the authors should add a part in which comment on the limitations of the study they performed and point out further research directions.

There are only minor aspects:

1.- Line 67: There are no examples of users’ preferences, only three dots.

2.- Line 92: Correct the English in the phrase: “To our knowledge, no study interested to compare a human-human collaboration and a human-robot collaboration for the same task about the production performance or the operators’ health or quality of the interactions”.

3.- Line 95. Correct the English of the last paragraph.

4.- Line 104: Correct the English: “Thirty-four volunteers participated to (in) the study,” I’m going to stop pointing out problems in the English.

7. PLOS authors have the option to publish the peer review history of their article (what does this mean?). If published, this will include your full peer review and any attached files.

Reviewer #2: No

Reviewer #3: No

Reviewer #4: No

---

## [Author Response · Author response to Decision Letter 1]

22 Jun 2023

- Comment 1:

Please review your reference list to ensure that it is complete and correct. If you have cited papers that have been retracted, please include the rationale for doing so in the manuscript text, or remove these references and replace them with relevant current references. If you need to cite a retracted article, indicate the article’s retracted status in the References list and also include a citation and full reference for the retraction notice.

Author’s comment:

We checked the reference list in the manuscript. We removed reference 32 (Richer N, Lajoie Y. Cognitive task modality influences postural control during quiet standing in healthy older adults. Aging Clin Exp Res. 2019;31: 1265–1270. doi:10.1007/s40520-018-1068-9), which was a duplicate of reference 19.

After checking the reference list, we did not find any retracted papers in this list. In your comment, you mentioned retracted papers. If you can provide us with the titles or authors of these papers, we will be able to modify or remove the corresponding references accordingly. Thank you in advance.

- Comment 2:

Any changes to the reference list should be mentioned in the rebuttal letter that accompanies your revised manuscript. 

Author’s comment:

We have not removed any papers from the initial manuscript. However, we added nine articles based on the comments in the previous review. Here are the nine new articles and references that have been cited:

o 3. Jain R, Meena ML, Sain MK, Dangayach GS. Impact of posture and upper-limb muscle activity on grip strength. International Journal of Occupational Safety and Ergonomics. 2019;25: 614–620. doi:10.1080/10803548.2018.1501972

o 4. Roman-Liu D, Tokarski T. Upper limb strength in relation to upper limb posture. International Journal of Industrial Ergonomics. 2005;35: 19–31. doi:10.1016/j.ergon.2004.07.002

o 11. Roveda L, Maggioni B, Marescotti E, Shahid AA, Maria Zanchettin A, Bemporad A, et al. Pairwise Preferences-Based Optimization of a Path-Based Velocity Planner in Robotic Sealing Tasks. IEEE Robot Autom Lett. 2021;6: 6632–6639. doi:10.1109/LRA.2021.3094479

o 17. Caroly S, Major M-E, Probst I, Molinié A-F. Le genre des troubles musculo-squelettiques: Interventions ergonomiques en France et au Canada. Travail Genre Sociétés. 2013;29: 49–67. doi:10.3917/tgs.029.0049

o 27. Baguley T. Understanding statistical power in the context of applied research. Applied Ergonomics. 2004;35: 73–80. doi:10.1016/j.apergo.2004.01.002

o 36. Rustum R, Adeloye AJ. Replacing Outliers and Missing Values from Activated Sludge Data Using Kohonen Self-Organizing Map. J Environ Eng. 2007;133: 909–916. doi:10.1061/(ASCE)0733-9372(2007)133:9(909)

o 54. Lamon E, De Franco A, Peternel L, Ajoudani A. A Capability-Aware Role Allocation Approach to Industrial Assembly Tasks. IEEE Robot Autom Lett. 2019;4: 3378–3385. doi:10.1109/LRA.2019.2926963

o 55. Bernar A, Prevot C, Legardeur J, Chanal H. Comment favoriser la transition vers l’industrie 4.0 dans le secteur textile-habillement : étude de cas du déploiement d’un cobot au sein de l’entreprise Petit Bateau. 18ème Colloque national S.mart. Carry le Rouet; 2023. 

o 56. Maurice P, Allienne L, Malaise A, Ivaldi S. Ethical and Social Considerations for the Introduction of Human-Centered Technologies at Work. 2018 IEEE Workshop on Advanced Robotics and its Social Impacts (ARSO). Genova, Italy: IEEE; 2018. pp. 131–138. doi:10.1109/ARSO.2018.8625830

o 57. Bao SS, Kapellusch JM, Merryweather AS, Thiese MS, Garg A, Hegmann KT, et al. Relationships between job organisational factors, biomechanical and psychosocial exposures. Ergonomics. 2016;59: 179–194. doi:10.1080/00140139.2015.1065347

Response to Reviewer #2

- Comment 1:

Reviewer’s comment:

The paper can be now accepted. All the concerns have been properly addressed in the revised paper and in the reply to review file.

Author’s comment:

Thank you for your comment.

Response to Reviewer #3

- Comment 1:

Reviewer’s comment:

Thanks for making changes. All the comments raised by reviewer are addressed by the authors in this version.

Author’s comment:

Thank you for your comment.

Response to Reviewer #4

- Comment 1:

Reviewer’s comment:

The authors should send the paper to a native editor to edit the English.

Author’s comment:

The paper has been submitted and was reviewed by a native American speaker to check and modify the English of the paper.

- Comment 2:

Reviewer’s comment:

In Table 1, it’s not clear what the authors mean by “a the condition with the highest value.”, “b the condition with lower value than a.”… This is not clear at all. The same is in table 2, 3 and 4.

Author’s comment:

To clarify the interpretation of the Tables, we have modified the coding to indicate the differences between the conditions.

We used the following three symbols (‡, § and #) with the companying legend: “‡, § and #: For a line, identical symbols indicate no difference between conditions, while different symbols indicate significant difference between conditions.” (example for Table 1 (p.15)).

- Comment 3:

Reviewer’s comment:

The added chapter “A theorical method to apply to a more ecological context” does add nothing significant to the paper and only points out one limitation. Instead, the authors should add a part in which comment on the limitations of the study they performed and point out further research directions.

Author’s comment:

We replaced this section by highlighting the main limitation of the study, which is that it was conducted in a laboratory setting. We conclude this section by discussing future research that would be valuable to extend the findings of this study in a more ecological situation (p.31-32).

- Comment 4:

Reviewer’s comment:

There are only minor aspects:

1.- Line 67: There are no examples of users’ preferences, only three dots.

2.- Line 92: Correct the English in the phrase: “To our knowledge, no study interested to compare a human-human collaboration and a human-robot collaboration for the same task about the production performance or the operators’ health or quality of the interactions”.

3.- Line 95. Correct the English of the last paragraph.

4.- Line 104: Correct the English: “Thirty-four volunteers participated to (in) the study,” I’m going to stop pointing out problems in the English.

Author’s comment:

We completed the space of the three dots by adding examples (p.4 l.68). All the English modifications were made with a proofreading by a native English speaker.

---

## [Decision Letter · Decision Letter 2]

27 Jul 2023

Does the introduction of a cobot change the productivity and posture of the operators in a collaborative task?

PONE-D-22-15982R2

Dear Kévin Bouillet

We’re pleased to inform you that your manuscript has been judged scientifically suitable for publication and will be formally accepted for publication once it meets all outstanding technical requirements.

Kind regards,

Pooya Sareh, PhD

Academic Editor

PLOS ONE

Reviewers' comments:

Reviewer's Responses to Questions

**Comments to the Author**

1. If the authors have adequately addressed your comments raised in a previous round of review and you feel that this manuscript is now acceptable for publication, you may indicate that here to bypass the “Comments to the Author” section, enter your conflict of interest statement in the “Confidential to Editor” section, and submit your "Accept" recommendation.

Reviewer #4: All comments have been addressed

2. Is the manuscript technically sound, and do the data support the conclusions?

Reviewer #4: Yes

3. Has the statistical analysis been performed appropriately and rigorously? 

Reviewer #4: Yes

4. Have the authors made all data underlying the findings in their manuscript fully available?

Reviewer #4: Yes

5. Is the manuscript presented in an intelligible fashion and written in standard English?

Reviewer #4: Yes

6. Review Comments to the Author

Reviewer #4: Thank you to the authors. They have addressed all my concerns and the paper is much better.

7. PLOS authors have the option to publish the peer review history of their article (what does this mean?). If published, this will include your full peer review and any attached files.

Reviewer #4: No

---

## [Editor Report · Acceptance letter]

31 Jul 2023

PONE-D-22-15982R2 

Does the introduction of a cobot change the productivity and posture of the operators in a collaborative task? 

Dear Dr. Bouillet:

I'm pleased to inform you that your manuscript has been deemed suitable for publication in PLOS ONE. Congratulations! Your manuscript is now with our production department. 

Kind regards, 

on behalf of

Dr. Pooya Sareh 

Academic Editor

PLOS ONE